

# Analysis of biogas production from sewage sludge combining BMP experimental assays and the ADM1 model

Mariana Erthal Rocha[1,2], Thais Carvalho Lazarino[1], Gabriel Oliveira[1], Lia Teixeira[2], Marcia Marques[2] and Norberto Mangiavacchi[1]

[1] Department of Mechanical Engineering, Rio de Janeiro State University, Rio de Janeiro, RJ, Brazil
[2] Department of Sanitary and Environmental Engineering, Rio de Janeiro State University, Rio de Janeiro, RJ, Brazil

## ABSTRACT

The Anaerobic Digestion Model No. 1 (ADM1) was employed to simulate methane ($CH_4$) production in an anaerobic reactor (AR), and the associated bench-scale biochemical methane potential (BMP) assay, having sewage sludge (SWS) from a municipal wastewater treatment plant (WWTP) as feedstock. The SWS presented the following physical-chemical characteristics: pH (7.4–7.6), alkalinity (2,382 $\pm$ 100 mg CaCO3 $L^{-1}$), tCOD (21,903 $\pm$ 1,000 mg $L^{-1}$), TOC (895 $\pm$ 100 mg $L^{-1}$), TS, TVS, and VSS (2.0%, 1.1%, and 0.8%, respectively). The BMP assay was conducted in six replicates under anaerobic mesophilic conditions (37 $\pm$ 0.1°C) for 11 days with a $CH_4$ yield registered of 137.6 $\pm$ 6.39 NmL $CH_4$ or 124 $\pm$ 6.72 $CH_4$ $g^{-1}$ $VS^{-1}$. When the results obtained with the BMP bench-scale reactors were compared to the output generated with computational data by the ADM1 model having as input data the same initial sewage tCOD, similar cumulative $CH_4$ production curves were obtained, indicating the accuracy of the ADM1 model. This approach allowed the characterization of the sludge and estimation of its biogas production potential. The combination of BMP assays, experimental data, and ADM1 model simulations provided a framework for studying anaerobic digestion (AD) processes.

# INTRODUCTION

Renewable energy plays a strategic role for energy security worldwide, since the main energy sources currently used are based on fossil fuels petroleum and coal, which are not renewable and pollute the environment (*Chen et al., 2018*; *Vasco-Correa et al., 2018*; *Sillero, Solera & Perez, 2022*).

Increasing amounts of sewage sludge (SWS) are generated all over the world at wastewater treatment plants (WWTPs) and its sustainable management is an important issue due to economic, environmental, and human health concerns (*Zan et al., 2022*; *Ma et al., 2019*).

Corresponding authors
Mariana Erthal Rocha,
marianaerthalrocha@gmail.com
Marcia Marques,
marciamarques@eng.uerj.br

Anaerobic digestion (AD) is a well-known complex microbial process in which organic waste, including SWS, is converted into bioenergy. This process integrates the set of waste management biotechnologies employing a diverse consortium of microorganisms (MO) to convert organic residues into $CH_4$-rich biogas (*Angelidaki et al., 2018*). AD of SWS is considered a relevant decarbonization process worldwide.

Producing and collecting $CH_4$ from SWS reduces water pollution and $CH_4$ emissions to the atmosphere (*Jiang et al., 2022*), meanwhile using this renewable source of energy contributes to saving environmental resources (*Deng et al., 2022*). However, the application of this approach in an industrial scale requires evaluation of biomass production and conversion systems, including feedstock selection and growth; harvest; storage conditions; bio-gasification; gas cleaning; gas use and residue processing (*Volschan Junior, de Almeida & Cammarota, 2021*).

The development of highly accurate mathematical models is currently a focal point in research efforts, as these models can effectively assess digester performance capabilities (*Achinas & Euverink, 2016*).

The current state-of-the-art model, the Anaerobic Digestion Model No. 1 (ADM1) was originally developed by the International Water Association—IWA Task Group (*Batstone et al., 2002*). The model was further improved resulting in the BSM2 version (*Alex et al., 2019*; *Donoso-Bravo et al., 2020*), the xp version, among others. One of the key points for successful application of mathematical models to describe a particular bioprocess is achieving adequate input characterization (*Batstone et al., 2002*). In this scenario extensive model calibration procedures are required to improve accuracy and sensitivity (*Weinrich & Nelles, 2021*). Mechanistic models have in common that they need to be carefully calibrated (*Sappl, Harders & Rauch, 2023*). Computational simulation (CS) evaluates the different processes and operation variables on the performance of the $CH_4$ biogas production in AD (*Jimenez et al., 2020*; *Batstone & Virdis, 2014*; *Batstone et al., 2015*). The ADM1 model has been used by several modelling studies in sewage AD (*Guo et al., 2023*; *García-Gen & Wouwer, 2021*; *Maharaj et al., 2019*; *Urtnowski-Morin et al., 2021*; *Zhao et al., 2019*).

The biochemical methane potential (BMP) assay has been widely used to test the anaerobic degradability of different organic wastes, and it is considered a suitable method to compare the degradability of different substrates (*Lavergne et al., 2018*). This test enables the assessment of the decomposability and the $CH_4$ conversion efficiency of diverse organic materials. *Owen et al. (1979)* initially introduced the BMP test, outlining a procedure to ascertain the decomposability of a substrate by monitoring the cumulative $CH_4$ production from an anaerobically incubated sample over a period of time (*Cabbai et al., 2013*; *Elbeshbishy, Nakhla & Hafez, 2012*; *Kafle & Chen, 2016*; *Raposo et al., 2011*; *VDI 4630, 2016*). However, the results obtained for the same substrate often differ among laboratories and much work to standardize such tests is still demanded (*Astals et al., 2013*; *Filer, Ding & Chang, 2019*; *Grosser, 2018*; *Raposo, Borja & Ibelli-Bianco, 2020*; *Strömberg, Nistor & Liu, 2014*). The German VDI 4630 (*VDI 4630, 2016*) is a BMP guideline that discusses the typical shape of methanogenic curves. However, the guideline presents only hypothetical idealized curves and does not discuss actual measurements, being not clear what might actually cause each response (*Koch et al., 2019*). According to *Filer, Ding*

**Table 1  BMP experiments operational parameters.**

| Operation | Unity | BMP assays |
|---|---|---|
| Temperature | °C | $37 \pm 0.1$ |
| Stir | – | Twice daily |
| Total volume | mL | 250 |
| Working volume | mL | 100 |
| Substrate | mL | Sewage sludge |
| Inoculum | mL | Sewage sludge |
| I/S ratio | v/v | 1:1 |

*& Chang (2019)* because of the lack of a standardized protocol, there have been serious drawbacks impacting the industry users, as the reliability of generated information could be questioned, due to the potential effects laboratory-specific experimental and operational conditions, as well as data presentation on the results, limiting the comparability of published results. During the years several authors have worked to improve the BMP method (*Angelidaki et al., 2009*; *Pearse, Hettiaratchi & Da Costa, 2020*; *Rodrigues et al., 2019*; *Wang et al., 2020*). Numerical prediction methods using initial data acquired from conventional bench-scale BMP assays were recently proposed (*Catenacci et al., 2022*; *Guo et al., 2023*; *Nabaterega, Nazyab & Eskicioglu, 2023*). However, the BMP method is still undergoing evolution, and researchers are investigating crucial aspects to further mitigate potential sources of bias.

In the present study, we developed a methodology that combines biochemical methane potential (BMP) assays and total chemical oxygen demand (tCOD) data with a computational simulator, employing the BSM2 version of the ADM1 model (*Alex et al., 2019*), for the coupled full-scale anaerobic reactor (AR) and the BMP assay systems. By doing so, we were able to generate a comprehensive and detailed view of the AR processes, including a description of the inflow and outflow SWS compositions. This methodology can be applied to analyze the biogas production and the underlying processes in full/real-scale applications.

## MATERIALS & METHODS

### Sampling and samples characterization

Samples of SWS were collected from an active anaerobic digester at a large municipal WWTP located in Rio de Janeiro, Brazil. The WWTP has a treatment capacity of 7,400 $m^3day^{-1}$ and a hydraulic retention time (HRT) of the AR is 28 days. It treats 2.5 $m^3s^{-1}$ of wastewater and serves a population equivalent to 1.5 million inhabitants. Standard methods (*APHA, 2017*) were used to measure physicochemical parameters.

The operational parameters applied to the BMP assay are shown in Table 1.

The initial and final pH values were measured using an MS Tecnopon model Mpa210 meter, and temperature levels were recorded with a digital thermometer. Chemical oxygen demand (COD) was determined using a Shimadzu UV-1800 UV–VIS Spectrophotometer, while alkalinity was measured *via* potentiometric titration. Total Solids (TS), total fixed

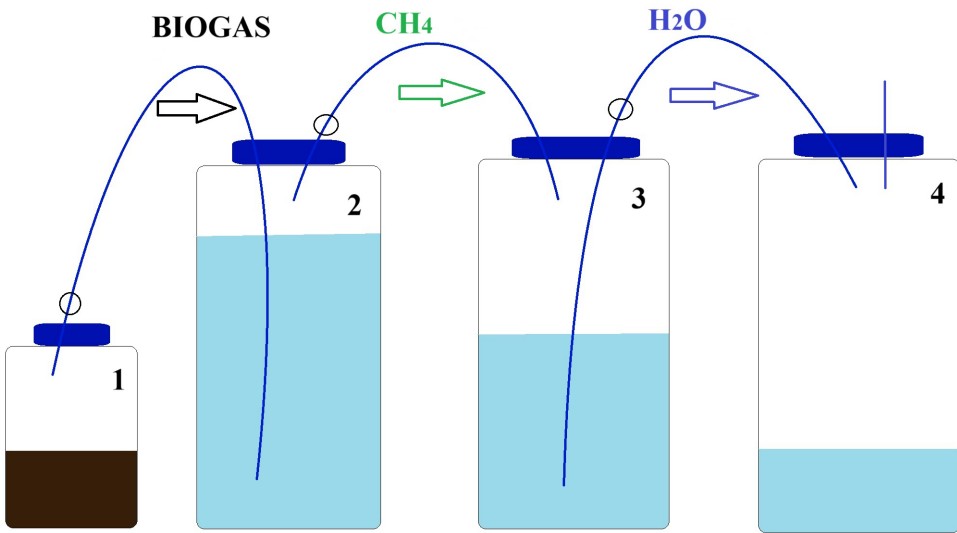

**Figure 1** **Schematic diagram of BMP apparatus used in BMP assay.** Vessel 1 is the BMP reactor, vessel 2 contains a 3 M NaOH scrubbing solution, vessel 3 contains 1 M NaCl saline solution. Vessel 4 is initially empty and collects the saline solution displaced by the $CH_4$ gas collected in vessel 3.

solids (TFS), total volatile solids (TVS), volatile suspended solids (VSS) were quantified using a gravimetric method with the analytical scale Gehaka AG200 (Gehaka, Sao Paolo, Brazil). Total organic carbon (TOC) analyses were carried out using a Shimadzu Total Organic Carbon Analyzer TOC 5000A (Shimadzu, Kyoto, Japan). Both inoculum and substrate, approximately 100 mL in total, were employed for the BMP assays with SWS as per the recommendation by *VDI 4630 (2016)*.

## BMP assays: experimental procedure

The BPM assay was carried out according to previous studies by *Angelidaki et al. (2009)* and the German Guideline for Fermentation of Organic Materials (*VDI 4630, 2016*), in order to evaluate $CH_4$ production from sewage sludge SWS in bench-scale.

The experiment was conducted in six replicates R1–R6 ($n = 6$) incubated during 11 days under mesophilic conditions ($37 \pm 0.1$ °C) using water-bath with digestion bottles of 250 mL total volume and 100 mL working volume (Fig. 1).

The BMP assay was carried out using four borosilicate Schott bottles (Fig. 1), chosen for their ability to withstand high temperatures and pressures. The first bottle in the sequence, referred to as the digester bottle (Fig. 1), featured pipes that had no contact with the inoculum, only with the gases produced. The second bottle was filled with a NaOH solution which served as a barrier to retain $CO_2$. The cannula was submerged in this basic solution as the gases generated in the digestion process were bubbled through it. The third bottle contained 1 L of a saturated NaCl solution and was equipped with a silicone stopper fitted with two 3-way taps. One tap was connected to the second bottle containing the alkaline solution, while the other tap was linked to the fourth bottle, which measured the displaced water volume.

To prevent gas leakage from the digesters, caps and connectors were securely sealed using high vacuum grease. In order to eliminate any existing $O_2$ in the system, $N_2$ gas was flushed into the headspace of the bottles for a period of 2 min. For bottle sealing, silicone stoppers with two holes each were employed. Two 3-way faucets were then inserted into the tops of these silicone stoppers, enabling external connections to the system. The analysis of gas volume produced during the experiment continued until the cumulative gas curve reached a plateau.

The sealing of the biodigester system was checked using a high-pressure pump, a differential dual port piezoresistive pressure transducer MPX5050DP, a Fluke multimeter, and an Arduino data logger. The bottles were gently shaken manually every day to prevent particle retention and system clogging.

This experimental setup allowed for precise measurements and ensured the integrity of the gas samples throughout the BMP assay.

## ADM1 model

The Anaerobic Digestion Model 1 (ADM1) is the quasi-industry standard for modelling the AD processes that result in $CH_4$ production from wastewater (*Allen et al., 2023*). It was developed by the International Water Association (IWA) and is based on several simpler AD models. It considers a continuously stirred tank reactor containing wastewater and 12 different bacterial groups that consume/produce 12 different substrates. This scenario is described by a system of 24 ordinary differential equations (ODEs). The model also considers physio-chemical reactions within the substrate itself, increasing the number of substrates to 23 and increasing the total number of state variables to 35. These physio-chemical reactions are classified into two categories: acid–base reactions and liquid–gas exchange, both of which can be modelled by ODEs, resulting in the total system being described by 35 ODEs.

In this work, the Benchmark Simulation Model 2 (BSM2), (*Alex et al., 2019*), which is a modified version of the IWA ADM1 model (*Batstone et al., 2002*), is employed. The model comprises two extracellular stages, namely disintegration and hydrolysis, and three intracellular stages: acidogenesis or fermentation, acetogenesis, and methanogenesis. The extracellular stages were modeled using first-order kinetics. In this work we will use, interchangeably, the variables and model parameters naming notation employed by *Alex et al. (2019)*, and *Batstone et al. (2002)*, and *Allen et al. (2023)*. For instance, complex composite concentration will be represented interchangeably by $X_c$ or X_c.

Initially, X_c was disintegrated into X_ch, X_pr, and X_li, as well as X_i and S_i, using a disintegration coefficient (kdis). Monod-type kinetics were used to describe substrate uptake in the intracellular biochemical reactions. In this study, two sequentially coupled simulators based on the ADM1 model were employed for the AR process and the BMP assay to provide a more comprehensive assessment of the inflowing sewage sludge, which will be explained in the following section.

In this study, we utilize a novel DAE-based (Differential Algebraic Equation) implementation of ADM1, developed in the Julia programming language. This implementation bears resemblance to a recent work by *Allen et al. (2023)*. Our

implementation, however, adopts the DAE approach, instead of the ODE approach employed by *Allen et al. (2023)*. Our choice of this approach was motivated by its exceptional computational efficiency, surpassing that of alternative DAE-based implementations in Java and Python. As a result, it empowers us to conduct intricate optimization analyses without incurring undue computational expenses.

The stoichiometric, biochemical and physicochemical model parameters values were those presented in *Rosén & Jeppsson (2005)*, and were not modified to represent the situation under study.

### Anaerobic reactor system simulation module

The first application of ADM1 was employed to simulate the treatment of influent sludge in the anaerobic reactor (AR), which is a continuous process. The SWS inflow is a complex particulate mixture, which is characterized by the COD mass (X_c). The primary constituents of X_c were found to be organic compounds (X_ch, X_pr, and X_li) and inorganic components (Xi and soluble inert material (Si)). The unity (kg COD m$^{-3}$) was utilized to input X_c and other organic substrates into the model. Subsequently, X_c was decomposed into five distinct components, according to their corresponding proportions, as determined by the yield of the product to the substrate (f_product_substrate), which led to hydrolysis, acidogenesis, acetogenesis, and methanogenesis. The input state variables of the soluble fractions (Ss) in the AR ADM1 module were set based on a best fit procedure as explained below.

### BMP system simulation module

The second application of ADM1 was employed to simulate the BMP, which is a batch assay. Inhibition was not applied, since the BMP tests had parameters such as temperature controlled, and pH maintained around the neutral value. The input parameters required for the BMP ADM1 module are the complete set of components that characterizes the AR outflow SW. The module predicts the curve of $CH_4$ production as a function of time as output.

### Combined AR and BMP ADM1 model parameters selection and validation

A methodology employing two coupled stages—AR and BMP—was applied to model the entire AD process, including the original bioreactor, and to infer additional parameters from BMP assays. A block diagram of the combined ADM1 model is shown in Fig. 2.

The first stage (AR ADM1 module) simulates a continuous-flow stirred-tank reactor (CSTR), while the second stage (BMP ADM1 module) simulates a batch reactor—the BMP assay. The output of the first module serves as the input for the second module. The input parameters of the first module are obtained through an iterative optimization solver. The solver minimizes the discrepancy between the simulated and measured values for the $CH_4$ production curve at selected times and the COD at the beginning of the BMP assay.

The methodology employed for determining the model parameters follows a least-squares optimization approach, elaborated as follows.
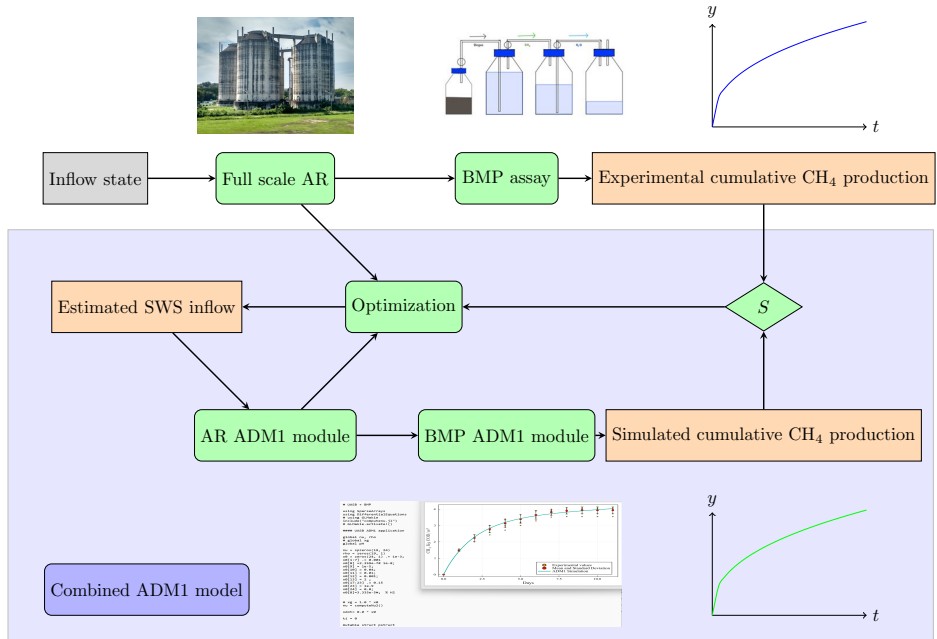

**Figure 2 Schematic diagram of the methodological approach adopted in the present study: combined BMP assay—ADM1 model.** Top: Experimental assay. The full scale AR receives SWS with Inflow state, and its outflow feeds the BMP assay that produces the experimental cumulative $CH_4$ production. Bottom: Combined ADM1 model, implemented in Julia language. An estimated SWS inflow enters the AR ADM1 module, and its output feeds the BMP ADM1 module, resulting in the Simulated cumulative $CH_4$ production. The sum of squared residuals S between the experimental and the simulated $CH_4$ production curves feeds the optimization routine that corrects the estimated SWS inflow parameters, until the sum of the squared residuals S is minimized.

The primary objective of the least-squares optimization is to identify the model parameters that most effectively align with the experimental dataset. This dataset comprises 'n' data points represented as pairs $(x_i, y_i), i = 1, \ldots, n$, where $x_i$ signifies an independent ADM1 parameter, and $y_i$ corresponds to a dependent parameter derived from the BMP assays. The model function is denoted as $f(x_i, \Pi)$, with m adjustable parameters encompassed in the parameter vector $\Pi$. The ultimate aim is to determine the parameter values for the model that offer the closest fit to the data. The fitting quality of a model to a given data point is quantified by the residual $r_i$, which signifies the disparity between the observed value of the dependent variable and the value predicted by the model:

$$r_i = y_i - f(x_i, \Pi)$$

The parameters $\Pi$ were determined through the weighted least-squares method to identify optimal values that minimize the sum of squared residuals, S, as in *Poggio et al. (2016)*:

$$S = \sum_{i=I}^{n} \frac{r_i^2}{\sigma_{m,i}}$$

Here, $y_{m,i}$ represents the ith measured value of the target measurement, assumed to be a normally distributed random variable. $f(x_i, \Pi)$ denotes the model prediction at the time corresponding to data point i, treated as a function of the set of parameters $\Pi$ to be estimated. Additionally, $\sigma_{m,i}$ stands for the standard error of the measurement, $y_{m,i}$, and serves as a weight for each term in the sum.

The standard error was estimated from the measured values, using the expression $\sigma_{m,i} = \frac{\sigma}{\sqrt{n}}$ where $\sigma$ is the standard deviation, and $n$ is the number of samples. In the case of the combined AR-BMP ADM1 model, the target measurement is the accumulated methane production along the BMP tests. The standard error of each measurement was estimated using the standard deviation of the BMP results.

Numerous optimization libraries are readily accessible within the Julia platform, facilitating rapid prototyping and experimentation with diverse strategies to address implementation optimization challenges. Various alternatives underwent testing, with Optimization.jl standing out. This library aims to amalgamate an array of optimization packages, both local and global, into a cohesive Julia interface. Optimization.jl introduces high-level attributes, such as seamless integration with automatic differentiation, rendering its utilization straightforward for most scenarios, all while retaining the entirety of options within a unified interface.

Though a simple gradient-based approach might suffice in certain instances, complications can arise due to positivity constraints. Gradient based methods suffer from many shortcomings such as slow convergence, difficulty with discontinuous problems, local minima and saddle points, among others. Consequently, an extensive array of methods available in the Optimization.jl library underwent evaluation in this work, ultimately revealing the DE/rand/1/bin method as the most suitable contender.

The "DE/rand/1/bin" optimization method is a specific variant of the Differential Evolution (DE) algorithm, a versatile technique for solving optimization challenges across domains.

The DE/rand/1/bin implemented in the Julia language Optimization.jl library was therefore used as the minimization technique, with a tolerance for convergence of 4E-3 in the objective function. The estimation process is repeated using different initial guesses of parameters to check the convergence of the algorithm towards the same optimum values.

The combined AR-BMP model employs, as input data, the COD values of the wastewater treatment plant (WWTP) sludge and the $CH_4$ production from the BMP assay. The iterative optimization method is employed to find the initial condition of the WWTP sludge (X_ch, X_pr, and X_li), as well as the effective hydraulic retention time (HRT) of the AR, which are input parameters of the AR ADM1 model. The search method is iterated until the initial conditions of the BMP ADM1 module, obtained from the AR module, result in a simulated $CH_4$ production curve that best matches the values at the points of the methanogenic curve obtained in the BMP assay.

As a result, the coupled AR-BMP model produces a simulated BMP methanogenic curve that best fits the values of the methanogenic curve obtained in the BMP experimental assays, in the least squares sense. The model was thus calibrated using the experimental

**Table 2  Sewage sludge physicochemical parameters.**

| Parameters | Unity | Anaerobic sewage |
|---|---|---|
| $pH_i$–$pH_f$ | – | 7.43–7.60 |
| TS | % | 2.0 |
| TVS | % | 1.1 |
| VSS | % | 0.8 |
| CODt | mg L$^{-1}$ | 21,903 $\pm$ 1000 |
| Alkalinity | mg CaCO$_3$L$^{-1}$ | 2,382 $\pm$ 100 |
| TOC | mg L$^{-1}$ | 895 $\pm$ 100 |

**Notes.**
pHi, pH initial; pHf, Final pH average; %TS, Total solids percentage; %TVS, Total volatile solids percentage; %VSS, Volatile suspended solids percentage; TCODi, Total initial Chemical Oxygen Demand; TOC, Total Organic Carbon.

data to accurately simulate the AD process. The output of the first model results in sludge with COD values similar to the WWTP biodigester sludge. Additionally, it produces a more complete description of the AR outflow SW which is consistent with the input data.

# RESULTS

## Substrate characterization

Table 2 shows the SWS physicochemical characterization used as feedstock in the BMP assays. Discrete changes in pH (7.43 $\pm$ 0.1 to 7.60 $\pm$ 0.1) were observed after 11 days in the reactor (Table 2), which is within the expected range, due to the growth of microorganisms (MO) and biogas production in all reactors without the addition of a buffer solution. The substrate pH was recorded at the beginning and end of the experiment to ensure that inhibition did not occur in the microbial communities. Alkalinity (2,382 $\pm$ 100 mg CaCO$_3$ L$^{-1}$) and CODt value (21,903 $\pm$ 1,000 mg L$^{-1}$) (Table 2) indicate the sludge's capability of buffering the reaction and the oxygen consumption capacity during the oxidation of the sludge organic matter (OM) into CO$_2$ and water. TOC (895 $\pm$ 100 mg CaCO$_3$ L$^{-1}$) is the carbon in the OM oxidized and measured through the release as CO$_2$. The biological processes lie in the ability of MO to use biodegradable organic compounds and transform them into by-products (*Lucas et al., 2015*). Regarding solid series, the following results were obtained after SWS characterization (Table 2): total solids (TS) 2.0%, total volatile solids (TVS) 1.1%, and volatile suspended solids (VSS) 0.8%. This suggests that anaerobic sludge contains more organic than inorganic compounds. The amount of biomethane yield produced is related to the mass of VS in these samples.

## BMP assay results

The curves of cumulative CH$_4$ production (Fig. 3) are within a narrow range of values and the standard deviation curve (Fig. 4) shows that the BMP reactors achieved good reproducibility. The round of experiments (R1 to R6) produced the following CH$_4$ values (Table 3): 124, 143, 137, 142, 140, and 130 NmL CH$_4$. These results indicate that the BMP assays were accurate and effective. During BMP assays, biogas production curves can follow diverse patterns (*Batstone et al., 2015*), and these patterns have meaningful implications (*Labatut, Angenent & Scott, 2011*). Temperature plays a crucial

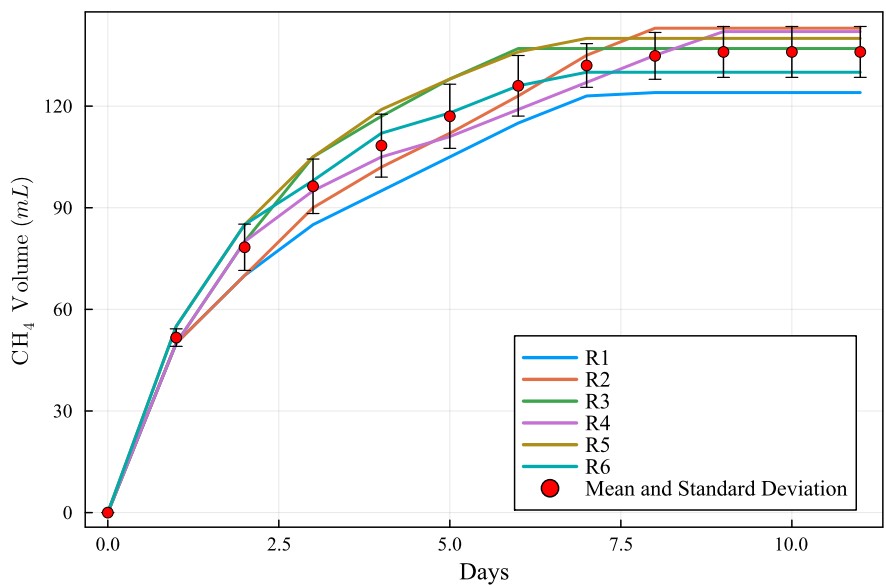

**Figure 3** Methane cumulative production during the experiment: R1 to R6 (124, 143, 137, 142, 140 and 130 NmL CH$_4$, respectively), mean and standard deviation.

role in microbial interactions and affects the stability and performance of AD and the thermodynamic equilibrium of biochemical reactions in the AD process (*Lin et al., 2017*).

The proposed BMP assay setup enabled daily measurement of biogas volume, which yielded satisfactory results. The alkaline solution used in the setup efficiently retained the produced $CO_2$. Furthermore, the BMP assays were conducted at mesophilic temperature, which contributed to the stability of the system. Mesophilic reactors are preferred for easy biodegradable biomass (*Issah, Kabera & Kemausuor, 2020*). The CH$_4$ yield production was found to be 113, 130, 124, 129, 127, and 118 mL CH$_4$ g$^{-1}$ VS$^{-1}$ in R1 to R6, respectively. These results demonstrate the effectiveness and accuracy of the experimental rounds conducted for the BMP assays (Table 4 and Fig. 5).

Table 5 shows the final (tCOD$_f$) values and the biodegradability rate (23–28%). The CH$_4$ yield can be normalized either per volume of substrate (mL CH$_4$ L$^{-1}$), substrate mass volatile solids (mL CH$_4$ g$^{-1}$ VS), or substrate mass chemical oxygen demand (COD) (mL CH$_4$ g$^{-1}$ COD$_{sub}$). The last method allows direct conversion of the results into percentage of OM converted to methane using the theoretical calculation of 0.350 m$^3$ CH$_4$ per kg COD converted (*McCarty, 1964*).

## Combined ADM1 computational simulation

The simulation results, employing the above data and model parameters presented in Table 6, are shown in Fig. 5, along with the results from the BMP assay. The simulated production curve is very similar to the one obtained in the BMP assay, indicating that the combined ADM1 model can reproduce the expected results. This makes the computational simulation (CS) a useful tool for elaborating and planning laboratory experiments. Tables 7

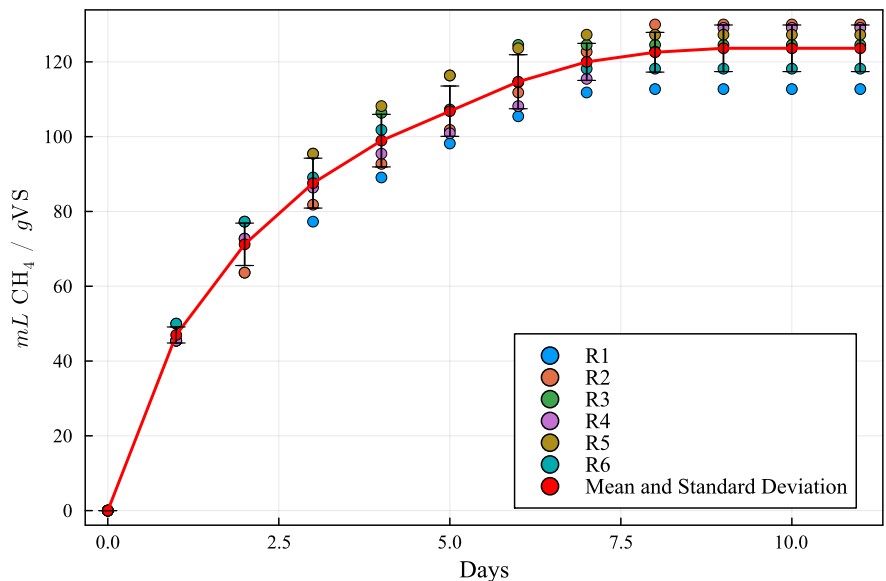

**Figure 4** Methane production yields R1 to R6 (113, 130, 124, 129, 127 and 118 mL CH$_4$/g VS, respectively), mean and standard deviation.

**Table 3** Methane production from sewage sludge in BMP assays ($n = 6$).

| Experiment | Unit | R1 | R2 | R3 | R4 | R5 | R6 | Mean ± s.d. |
|---|---|---|---|---|---|---|---|---|
| CH$_4$ | mL | 124 | 143 | 137 | 142 | 140 | 130 | 136 ± 7.5 |
| CH$_4$ yield | mL CH$_4$ g$^{-1}$ VS | 113 | 130 | 124 | 129 | 127 | 118 | 124 ± 6.23 |
| CH$_4$ yield | mL CH$_4$ g$^{-1}$ COD | 5.6 | 6.5 | 6.2 | 6.1 | 5.9 | 6.3 | 6.15 ± 0.9 |

and 8 show the SWS composition according to the ADM1 model, applying the iterative optimization method and based on the SWS COD initial real value in the BMP assay feedstock.

The data in Table 7 show that the AR inflow is well characterized basically by a mixture of complex composites (40.769 kg COD m$^{-3}$), with a small addition of protein (8.127 kg COD m$^{-3}$), and trace amounts of additional lipids (8.127 kg COD m$^{-3}$) and inert particulate (0.087 kg COD m$^{-3}$).

It must be stressed that the complex composites category in ADM1 results in a predefined distribution of carbohydrates, proteins, lipids, and inert particulate matter when hydrolyzed. Thus, the net effect of including additional protein, lipids and inert particulate is to correct the default composition of the complex composite and to produce simulation results that better approximate the observed data.

The results obtained by the ADM1 modeling were very close to those obtained experimentally in the BMP test, illustrating the quality of the model proposed by the International Water Association (IWA). By using this model together with a least-squares

**Table 4  Methane production from sewage sludge in experimental assays in the literature.**

| Reactor type | OS | Temp (°C) | HRT (days) | Ym (mL CH$_4$ g$^{-1}$ VS) | Reference |
|---|---|---|---|---|---|
| Batch BMP | PS | 37 | 20 | 138.2 | *Alves et al. (2020)* |
| Batch | SS | 35 | n.a. | 182 | *Bai & Chen (2020)* |
| Batch BMP | SS | 37 | n.a. | 124.4 | *Pan et al. (2019)* |
| Batch BMP | SS | 35 | n.a. | 182 | *Ripoll et al. (2020)* |
| BMP | PS | 35 | 13 | 159 | *Xie, Wickham & Nghiem (2017)* |
| BMP | SS | m | 35 | 121 | *Kashi et al. (2017)* |
| Batch | SS | 35 | 40 | 88.1 | *Zou et al. (2018)* |
| Batch | SS | 35 | 10 | 135.6 | *Xie et al. (2020)* |
| Batch | SS | 37 | 35 | 142.7 | *Zhang et al. (2014)* |
| Batch | SS | 37 | 11 | 124 | This study |

**Notes.**

OS, organic substrate; SS, sewage sludge; PS, primary sludge; m, mesophilic; Ym, CH$_4$ yield; n.a., not available.

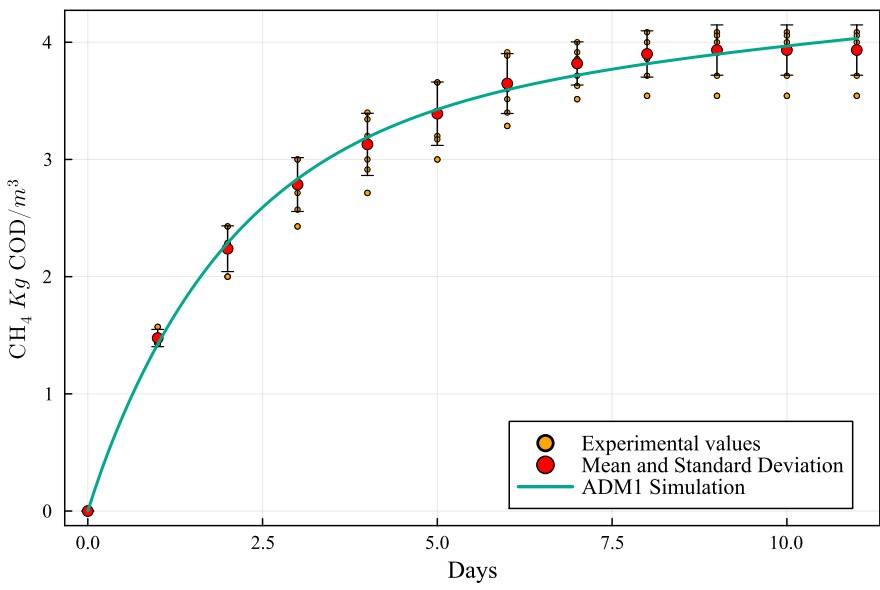

**Figure 5  Methanogenic curve obtained with the BMP experimental assay *versus* the curve obtained with the computational simulation using the ADM1 simulation model.**

iterative method, a detailed SWS composition consistent with the BMP data was found (Table 8).

Table 8 shows the main concentrations present in the AR sludge, which include inert particulate, complex composite, inert soluble, and amino acid consumers. The composition of the AR sludge indicates a significant decrease in the concentration of complex composite (to approximately 5 kg COD m$^{-3}$) and a considerable increase in the concentrations of inert particulate and inert soluble. Results in Table 8 were obtained by numerical calculation,

**Table 5  Biodegradability data in BMP assays ($n = 6$).**

| Replicates | pH$_{final}$ | TCOD$_f$ (mg L$^{-1}$) | COD removal (%) |
|---|---|---|---|
| R1 | 7.5 | 16,820 | 23.2 |
| R2 | 7.65 | 15,680 | 28.4 |
| R3 | 7.61 | 16,714 | 23.7 |
| R4 | 7.6 | 16,316 | 25.5 |
| R5 | 7.58 | 16,512 | 24.7 |
| R6 | 7.75 | 16,970 | 23.5 |

and refer to the modeled composition output of the AD, and feed to the BMP model that best approximates the results of experimental BMP results.

The low concentration of the composite X_c when compared to the inert X_i and S_i concentrations is consistent with a composition of the output variable values of a AD, where the Xc undergoes an efficient consumption.

Also the high values of X_aa and X_ac, when compared to other five biomass fractions, are compatible with results found in other AD simulations and experimental results.

Total acetate concentration is fed by six processes (with biochemical process rates $\rho_5$ to $\rho_{10}$) thus resulting in an acetate concentration that is comparatively higher than each of the six original sources. Also, acetate consumers do not have a faster metabolism compared to other consumers in the process. For instance, K_m_ac = 8 d$^{-1}$, while K_m_pro = 6 d$^{-1}$. Combining these two factors (larger concentration of acetate and relatively slow acetate consumers biochemical process rates), a larger mass of acetate consumers X_ac is required to consume the results of the acetogenic metabolic pathway production. If not properly taken into account, the high concentration of acetate could lead to reduced methane production due to the inhibition of acetoclastic methanogens caused by low pH.

On the other hand, protein load in the feed is relatively high, and amino acid production is proportionally high, thus a larger mass of amino acid consumers X_aa is to be found at the AD process output when the process reaches a steady state. A relatively high concentration of proteins, relative to carbohydrates and lipids is not uncommon in wastewaters in tropical climates.

By matching the ADM1 results with the BMP assays experimental measurements, an equivalent composition was obtained (Table 8 and Fig. 5). Table 9 provides a numerical comparison between BMP CH$_4$ production experimental results and ADM1 computational results, showing a maximum discrepancy of less than 2.8%. Based on the similarity of the results, it can be concluded that the model is suitable to characterize SWS and biogas production potential.

## Results of validation of ADM1 model DAE-Based Julia implementation

The new Julia language implementation of the DAE-ADM1 model is validated comparing results with the ODE-Based Julia implementation (*Allen et al., 2023*). Validation tests were performed on various benchmark problems, showing very good agreement. In this section

**Table 6  ADM1 stoichiometric, biochemical and physiochemical model parameters values used in the simulations.** Values not marked are taken from Rosen and Jeppson (2006). Values highlighted with "*" are modified, and the original values are shown between parentheses "( )".

| Parameter | Value | Unit | Parameter | Value | Unit | Parameter | Value | Unit |
|---|---|---|---|---|---|---|---|---|
| R | 0.083145 | bar / M K | f_ac_aa | 0.4 | – | pH_UL_h2 | 6 | – |
| T_base | 298.15 | K | C_va | 0.024 | kmole C/Kg COD | pH_LL_h2 | 5 | |
| P_atm | 1.013 | bar | Y_aa | 0.08 | | k_dec_X_su | 0.02 | |
| T_op | 308.15 | K | Y_fa | 0.06 | | k_dec_X_aa | 0.02 | |
| f_sI_xc | 0.1 | | Y_c4 | 0.06 | – | k_dec_X_fa | 0.02 | |
| f_xI_xc | 0.2 | | Y_pro | 0.04 | | k_dec_X_c4 | 0.02 | 1/d |
| f_ch_xc | 0.2 | | C_ch4 | 0.0156 | kmole C/Kg COD | k_dec_X_pro | 0.02 | |
| f_pr_xc | 0.2 | – | Y_ac | 0.05 | – | k_dec_X_ac | 0.02 | |
| f_li_xc | 0.3 | | Y_h2 | 0.06 | – | k_dec_X_h2 | 0.02 | |
| N_xc | 0.002685714 | | k_dis | 0.5 | | T_ad | 308.15 | K |
| N_I | 0.004286 | kmole N/Kg COD | k_hyd_ch | 10 | | K_h2o | 2.08E−14 | |
| N_aa | 0.007 | | k_hyd_pr | 10 | 1/d | K_a_va | 1.38E−05 | |
| C_xc | 0.02786 | | k_hyd_li | 10 | | K_a_bu | 1.51E−05 | |
| C_sI | 0.03 | | K_S_IN | 0.0001 | M | K_a_pro | 1.32E−05 | M |
| C_ch | 0.0313 | | k_m_su | 30 | 1/d | K_a_ac | 1.74E−05 | |
| C_pr | 0.03 | | K_S_su | 0.5 | Kg COD/m³ | K_a_co2 | 4.94E−07 | |
| C_li | 0.022 | kmole C/Kg COD | pH_UL_aa | 5.5 | | K_a_IN | 1.11E−09 | |
| C_xI | 0.03 | | pH_LL_aa | 4 | – | k_AB_va | 1.00E+10 | |
| C_su | 0.0313 | | k_m_aa | 50 | 1/d | k_AB_bu | 1.00E+10 | |
| C_aa | 0.03 | | K_S_aa | 0.3 | Kg COD/m³ | k_AB_pro | 1.00E+10 | |
| f_fa_li | 0.95 | – | k_m_fa | 6 | 1/d | k_AB_ac | 1.00E+10 | 1/M d |
| C_fa | 0.0217 | kmole C/Kg COD | K_S_fa | 0.4 | Kg COD/m³ | k_AB_co2 | 1.00E+10 | |
| f_h2_su | 0.19 | | K_I_h2_fa | * 5.00E−04 (5.00E−06) | | k_AB_IN | 1.00E+10 | |
| f_bu_su | 0.13 | | k_m_c4 | 20 | 1/d | p_gas_h2o | 0.055667745 | bar |
| f_pro_su | 0.27 | – | K_S_c4 | 0.2 | Kg COD/m³ | k_p | 50000 | m³/d bar |
| f_ac_su | 0.41 | | K_I_h2_c4 | 1.00E−05 | | k_L_a | 200 | 1/d |
| N_bac | 0.005714286 | kmole N/Kg COD | k_m_pro | 13 | 1/d | K_H_co2 | 0.027146693 | |
| C_bu | 0.025 | | K_S_pro | 0.1 | Kg COD/m³ | K_H_ch4 | 0.001161903 | M/bar |
| C_pro | 0.0268 | | K_I_h2_pro | 3.50E−06 | | K_H_h2 | 0.000738465 | |
| C_ac | 0.0313 | kmole C/Kg COD | k_m_ac | 8 | 1/d | V_liq | * 3485.4 (3400) | |
| C_bac | 0.0313 | | K_S_ac | 0.15 | Kg COD/m³ | V_gas | 300 | m³ |
| Y_su | 0.1 | | K_I_nh3 | 0.0018 | M | Q_ad | * e246.67 (170) | m³/d |
| f_h2_aa | 0.06 | | pH_UL_ac | 7 | | tresX | 40 | d |
| f_va_aa | 0.23 | | pH_LL_ac | * 5.9 (6) | - | k_dec_all | 0.02 | 1/d |
| f_bu_aa | 0.26 | – | k_m_h2 | 35 | 1/d | | | |
| f_pro_aa | 0.05 | | K_S_h2 | 7.00E−06 | Kg COD/m³ | | | |

we present the results of the test performed on the data obtained in the previous section (Tables 6 and 7), in order to provide a validation of the results of the simulations performed in this work.

Figures 6 to 10 show the results of the AR simulation using the current DAE implementation, compared to the results obtained by the DAE implementation of *Allen et al. (2023)*. The validation tests demonstrate that the results have a very good agreement,

**Table 7  ADM1 model input of the biodigester.** The values for complex composites, proteins, lipids, inert particulate, highlighted with "*", are determined by the optimization routine.

| Variables | Value | Unit |
|---|---|---|
| S_su | 0.01 | Kg COD/ m$^3$ |
| S_aa | 0.001 | Kg COD/ m$^3$ |
| S_fa | 0.001 | Kg COD/ m$^3$ |
| S_va | 0.001 | Kg COD/ m$^3$ |
| S_bu | 0.001 | Kg COD/ m$^3$ |
| S_pro | 0.001 | Kg COD/ m$^3$ |
| S_ac | 0.001 | Kg COD/ m$^3$ |
| S_h2 | 1.0E−8 | Kg COD/ m$^3$ |
| S_ch4 | 1.0E−5 | Kg COD/ m$^3$ |
| S_IC | 0.04 | kmole C/m$^3$ |
| S_IN | 0.01 | kmole N/m$^3$ |
| S_I | 0.02 | Kg COD/ m$^3$ |
| X_xc | * 40.769 | Kg COD/ m$^3$ |
| X_ch | 0.00 | Kg COD/ m$^3$ |
| X_pr | * 8.127 | Kg COD/ m$^3$ |
| X_li | *0.252 | Kg COD/ m$^3$ |
| X_su | 0.00 | Kg COD/ m$^3$ |
| X_aa | 0.01 | Kg COD/ m$^3$ |
| X_fa | 0.01 | Kg COD/ m$^3$ |
| X_c4 | 0.01 | Kg COD/ m$^3$ |
| X_pro | 0.01 | Kg COD/ m$^3$ |
| X_ac | 0.01 | Kg COD/ m$^3$ |
| X_h2 | 0.01 | Kg COD/ m$^3$ |
| X_I | * 0.087 | Kg COD/ m$^3$ |
| S_cat | 0.04 | kmole/m |
| S_an | 0.10 | kmole/m |

both for the steady state results and for the transient regime results, thus providing a quantitative validation of the new implementation.

The time-dependent curves (Figs. 6 to 10) show that, while some concentrations attain the steady state value very shortly after the startup (about 10 days, as it is the case of X_c), most concentrations only attain their steady state after a much longer period (typically 60 days, as in the case of X_su). However, it can be observed that the concentrations do not vary after 80 days, thus the current 90 day simulation period is sufficient for the AR to reach steady state conditions.

The computational time of the current DAE-based Julia implementation compares favorably with the ODE-Based Julia implementation by a large factor. The mean execution time for the AR simulation with the current DAE-based implementation is 22.15 ms, while with the ODE-based implementation (*Allen et al., 2023*), the mean execution time is 1,231 ms. Therefore, the current ODE-based implementation is approximately 56 times faster than the DAE-based implementation. Thus, the new implementation is more suitable to

**Table 8  ADM1 model AR output variables and BMP initial composition.**

| Sludge composition and model variables (Kg COD/m$^3$) | | | |
|---|---|---|---|
| $x0_1 = 1.5031e{-}02$ | S_su, Monosaccharide | $x0_{13} = 5.0183e{+}00$ | X_c, Complex Composite |
| $x0_2 = 6.7129e{-}03$ | S_aa, Amino Acid | $x0_{14} = 4.9515e{-}02$ | X_ch, Carbohydrate |
| $x0_3 = 1.2853e{-}01$ | S_fa, LCFA | $x0_{15} = 1.5778e{-}01$ | X_pr, Proteins |
| $x0_4 = 1.4532e{-}02$ | S_va, Total Valerate | $x0_{16} = 7.7605e{-}02$ | X_li, Lipids |
| $x0_5 = 1.6795e{-}02$ | S_bu, Total Butyrate | $x0_{17} = 6.0868e{-}01$ | X_su, Sugar Consumers |
| $x0_6 = 2.0244e{-}02$ | S_pro, Total Propionate | $x0_{18} = 1.4414e{+}00$ | X_aa, Amino Acid Consumers |
| $x0_7 = 4.2027e{-}02$ | S_ac, Total Acetate | $x0_{19} = 4.9932e{-}01$ | X_fa, LCFA Consumers |
| $x0_8 = 0.00$ | S_h2, Hydrogen Gas | $x0_{20} = 5.2954e{-}01$ | X_c4, Valerate/Butyrate Consum. |
| $x0_9 = 0.00$ | S_ch4, Methane Gas | $x0_{21} = 1.6868e{-}01$ | X_pro, Propionate Consumers |
| $x0_{10} = 6.3799e{-}01$ | S_IC, Inorganic Carbon | $x0_{22} = 1.0750e{+}00$ | X_ac, Consumers of Acetate |
| $x0_{11} = 1.2453e{-}01$ | S_IN, Inorganic Nitrogen | $x0_{23} = 4.5639e{-}01$ | X_h2, Hydrogen Consumers |
| $x0_{12} = 3.7066e{+}00$ | S_I, Inert Soluble | $x0_{24} = 7.5005e{+}00$ | X_I, Inert Particulate |

**Table 9  Comparison between BMP CH$_4$ production experimental results and ADM1 computational results (mL CH$_4$ g$^{-1}$ VS$^{-1}$).**

| Day | CH$_4$ BMP | CH$_4$ ADM1 | Error (%) |
|---|---|---|---|
| 0 | 0 | 0.0000 | 0 |
| 1 | 46.9697 | 46.4013 | −1.2101 |
| 2 | 71.2121 | 72.8888 | 2.3545 |
| 3 | 87.5758 | 89.5668 | 2.2735 |
| 4 | 98.4848 | 100.3918 | 1.9363 |
| 5 | 106.3636 | 107.6856 | 1.2429 |
| 6 | 114.5455 | 112.8359 | −1.4925 |
| 7 | 120.0000 | 116.6777 | −2.7685 |
| 8 | 122.5758 | 119.7153 | −2.3336 |
| 9 | 123.6364 | 122.2534 | −1.1186 |
| 10 | 123.6364 | 124.4768 | 0.6797 |
| 11 | 123.6364 | 126.4973 | 2.3140 |

perform model parameter estimations and operational parameters optimization. The tests were performed on an Intel(R) Core(TM) i7-1060NG7 CPU @ 1.20 GHz, with 16GB 3733 MHz LPDDR4X memory, macOS Ventura 13.5.2, and Julia v1.9.

## DISCUSSION

### Operational parameters and biogas production

AD pH stability is the main controlling factor in balancing the system (*Issah, Kabera & Kemausuor, 2020*). No pH adjustments were made during the BMP assays due to the buffer capacity of the SWS. The twice-daily stirring and agitation of the digesting vessels allow for the release of gas during the assays and at the same time avoiding clogging of the connecting valves. Biogas yield decreases when the pH value is higher than 7.6 or lower than 6.8 (*Pilarski et al., 2020*). Process variables, such as temperature and pH, are the principal

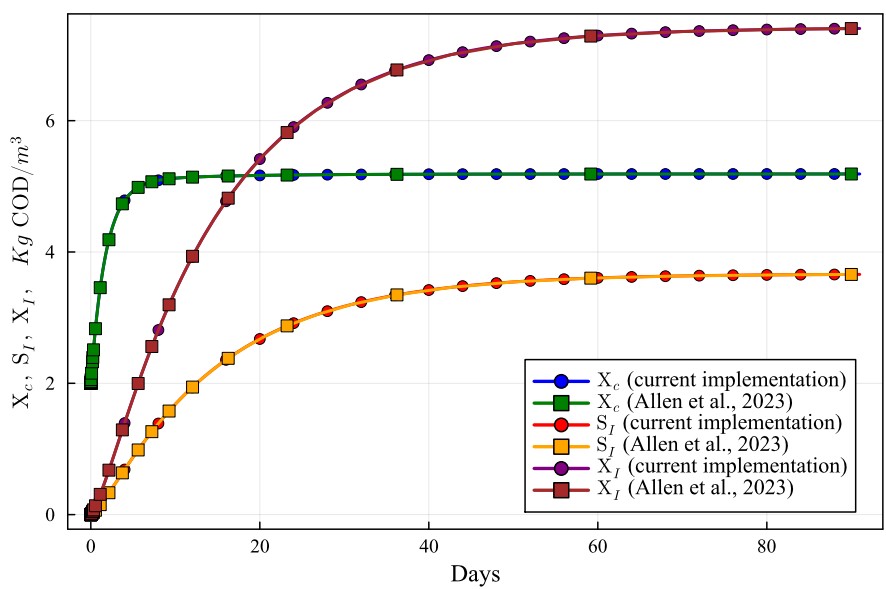

**Figure 6** Concentration curves of Xc, SI an XI obtained with the computational simulation of the AR using the current DAE-ADM1 model implementation, compared with the results obtained with ODE-ADM1 model (*Allen et al., 2023*). Lines with circles correspond to values obtained in the current simulations. Curves with squares are obtained with the ODE-ADM1 model, showing virtually identical results.

factors affecting the smooth digestion process (*Castellano-Hinojosa et al., 2018*). At very high or very low temperatures, bacterial and archaeal activities may be curtailed, leading to low yields, and unbalanced pH could result in volatile fatty acid (VFA) accumulation that could result in MO mortality (*Issah, Kabera & Kemausuor, 2020*), especially methanogenic ones.

High values for alkalinity indicate that the reaction is buffered, so the pH does not undergo major changes (Angelidaki et al., 2013). The SWS alkalinity value in this study (2,382 $\pm$100 mg CaCO$_3$ L$^{-1}$) is similar to the alkalinity found in other SWS from WWTP as observed by *Grosser et al. (2020)*, who investigated the BMP of SWS from a WWTP in Poland (2,823 mg CaCO$_3$ L$^{-1}$).

The tCOD value (21,903 $\pm$ 1,000 mg L$^{-1}$) is consistent with values found in other studies, such as 25,250 mg L$^{-1}$ by *Park et al. (2021)* and 22,300 mg L$^{-1}$ by *Wickham et al. (2018)*.

As MO converts chemical energy to CH$_4$, this is directly associated with the maximum energy that can be recovered as biogas (*Raposo et al., 2012*). The inoculum taken from an active anaerobic digester (AR) that is digesting complex organic matter (OM) and is at a steady state at the time of sampling provides a highly diverse microbial community, able to digest a large variety of organic molecules (*Holliger et al., 2016*).

Similar results for SWS mono-digestion (Table 4) were also reported by *Alves et al. (2020)* (138.2 mL CH$_4$ g$^{-1}$VS$^{-1}$), *Pan et al. (2019)* (124.43 mL g$^{-1}$ VS$^{-1}$), *Park et al. (2021)* (100–175 mL CH$_4$ g$^{-1}$ VS$^{-1}$), and *Zou et al. (2018)* (88.1 mL CH$_4$ g$^{-1}$ VS$^{-1}$).

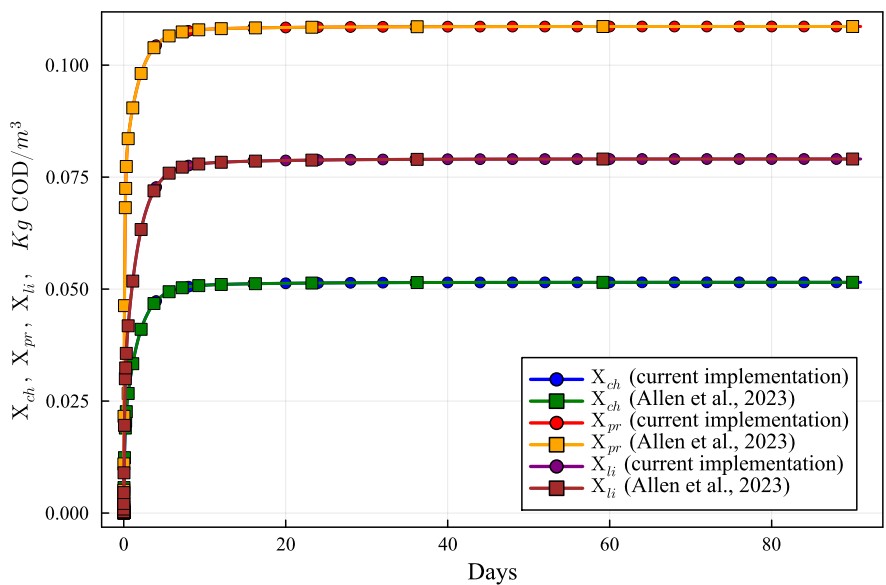

**Figure 7** **Concentration curves of Xch, Xpr and Xli obtained with the computational simulation of the AR using the current DAE-ADM1 model implementation, compared with the results obtained with ODE-ADM1 model (***Allen et al., 2023***).** Lines with circles correspond to values obtained in the current simulations. Curves with squares are obtained with the ODE-ADM1 model, showing virtually identical results.

The sanitation sector, with the use of SWS AD, has the possibility of transforming an environmental liability into an energy asset.

## Carbon content and sludge biodegradability

The biodegradability rate achieved in the present study is consistent with the COD removal percentages reported in previous studies based on SWS BMP assays, such as *Maragkaki et al. (2018)*, *Kashi et al. (2017)*, and *Zhang, Hu & Lee (2016)*, which achieved removal percentages of 28.9%, 16.0%, and 25.2%, respectively. OM is measured by the amount of carbon in a feedstock (*Ferguson, Coulon & Villa, 2018*; *Gohil et al., 2018*) and biomethane yield is affected by the VS content (*Mayer et al., 2014*). Therefore, $CH_4$ production is directly related to the degradation of VS (*Angelidaki et al., 2009*) being VS the OM component of TS.

Systems used in AD are classified according to the percentage of TS in the feedstock (*Yi et al., 2014*). The biogas yield mainly depends on the content of organic compounds in the feedstock, including fats, proteins, and carbohydrates, which are biologically degradable under AD (*Abdul Aziz, Hanafiah & Mohamed Ali, 2019*). Anaerobic MO can be inhibited by substances present in the substrate or by compounds generated in the metabolism itself (*Mustapha et al., 2018*). The biochemical methane production potential of the substrates intended for anaerobic digestion and their specific organic loads can be utilized to design various components of full-scale AD plants, including the size of digesters and the potential for utilizing the generated biogas (*Filer, Ding & Chang, 2019*).

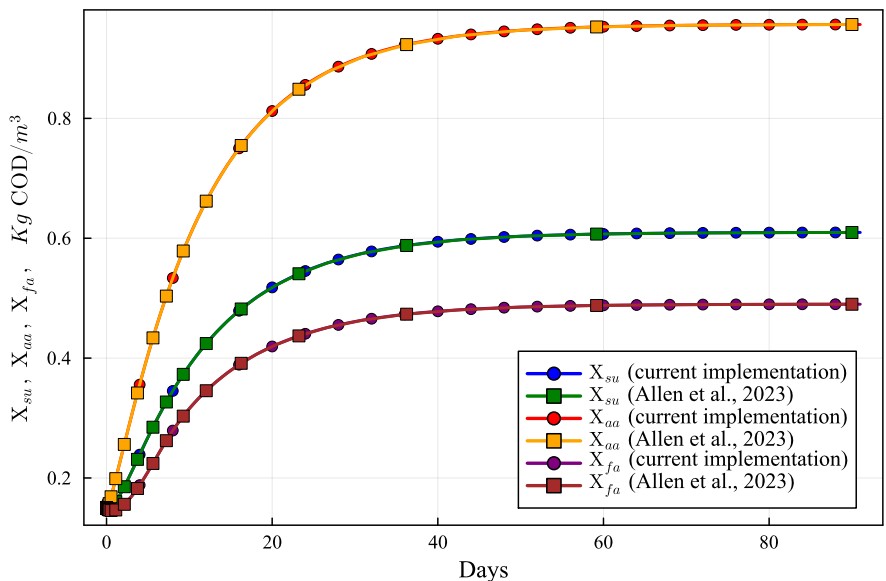

**Figure 8 Concentration curves of Xsu, Xaa and Xfa obtained with the computational simulation of the AR using the current DAE-ADM1 model implementation, compared with the results obtained with ODE-ADM1 model** (*Allen et al., 2023*). Lines with circles correspond to values obtained in the current simulations. Curves with squares are obtained with the ODE-ADM1 model, showing virtually identical results.

SWS is an optimal inoculum for BMP assays because of the diversity of its microbial population (*Raposo et al., 2011*). A well-functioning AD must contain a balanced microbial consortium community for efficient biogas production (*Issah, Kabera & Kemausuor, 2020*). The experimental results showed that the SWS obtained from a municipal WWTP has the capability to produce $CH_4$, and it is considered as a feasible strategy for bioenergy production. The biodegradability properties of substrates and production of inhibitory intermediate products will mainly control the kinetics of AD different steps and define the shape of the biogas production curve, identifying important characteristics of substrates and anticipating digestion issues (*Labatut, Angenent & Scott, 2011*).

COD is commonly used to measure the organic strength of liquid effluents. At WWTPs, each kilogram of COD removed will yield 0.35 m³ of $CH_4$ gas at standard temperature and pressure (*Jingura & Kamusoko, 2017*). The theoretical $CH_4$ yield can be calculated from the COD of a substrate, and biogas production in relation to COD is about 0.5 L g$^{-1}$ COD removed, corresponding to a $CH_4$ production of approximately 0.35 L g$^{-1}$ of COD removed (*Angelidaki & Sanders, 2004*).

### Feasibility of using the model as a tool to predict BMP results

The ADM1 CS can be used to compare the decrease in COD observed experimentally with the values obtained from numerical simulation. Furthermore, the model can be validated by comparing and calibrating the methanogenic activity and production curves. AD processes are conducted under laboratory conditions and scaled to pilot-scale trials.
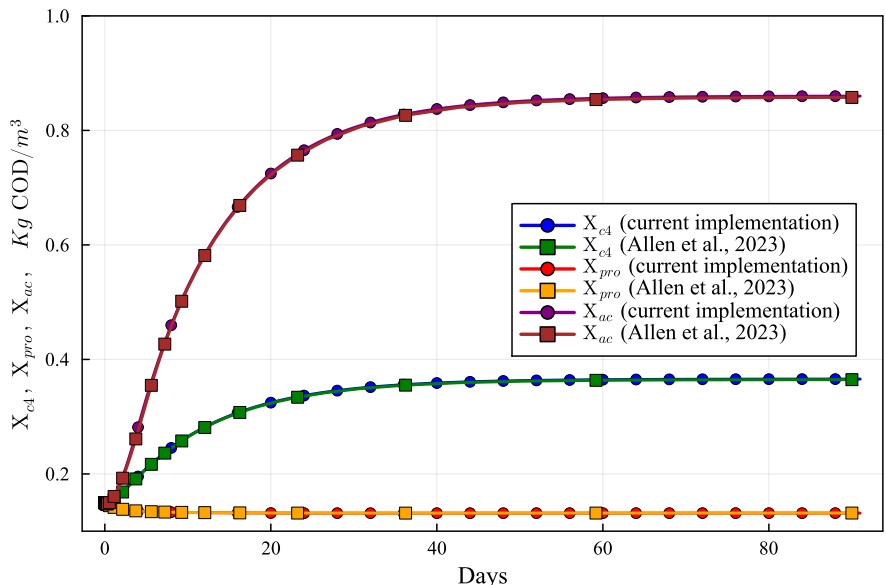

**Figure 9** **Concentration curves of Xc4, Xpro and Xac obtained with the computational simulation of the AR using the current DAE-ADM1 model implementation, compared with the results obtained with ODE-ADM1 model** (*Allen et al., 2023*). Lines with circles correspond to values obtained in the current simulations. Curves with squares are obtained with the ODE-ADM1 model, showing virtually identical results.

This methodology can be applied to support the development of experiments and full-scale reactor projects. The generated BMP data's scalability and transferability allow the results to be applied to larger-scale systems, facilitating the development and optimization of AD processes (*Jingura & Kamusoko, 2017*).

The ADM1 model can be adapted to individual cases. There are certain factors in ADM1 that require the user's discretion, such as the fractionation of composition (Xc) and the definition of inert ingredients, soluble (SI) and particulate (XI). However, there are hypotheses that may restrict the model's applicability in anoxic environments. For instance, the influent particulate composite substrate and the cytolysis product use the same component Xc, which requires a disintegration process before hydrolysis. Therefore, characterizing the influent sludge becomes particularly challenging.

## Combined AR-BMP model output and comparison with BMP assay: estimation of outflow composition, biogas production, and operation parameters of the full-scale AR

The methodology developed in this study involved using two coupled ADM1 models to validate the complete AD process, including the original bioreactor, and to infer additional parameters from the BMP assays. The first model simulates a continuous-flow stirred-tank reactor (CSTR), and the second model simulates the BMP assay, which is a batch reactor. The input of the second model is given by the output of the first model, and the input parameters of the first model are obtained by means of an iterative optimization method

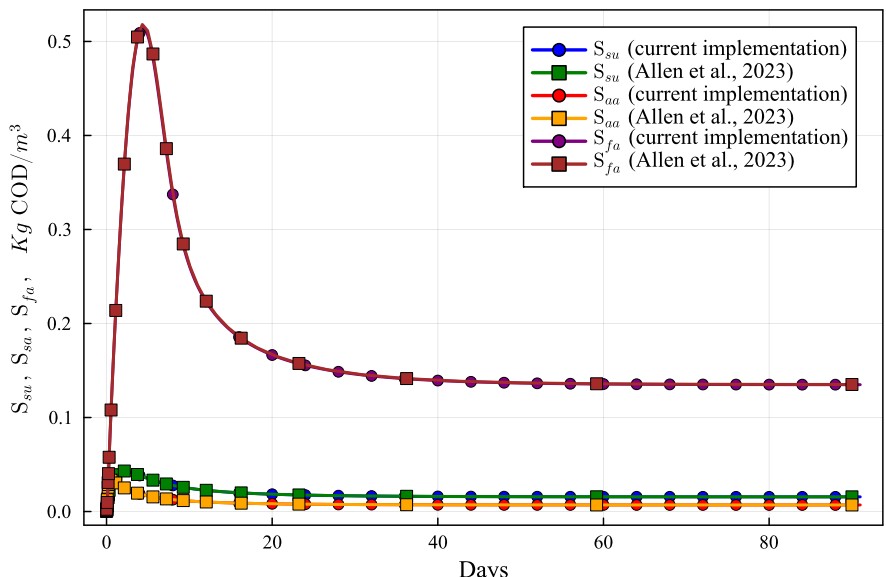

**Figure 10** **Concentration curves of Ssu, Saa and Sfa obtained with the computational simulation of the AR using the current DAE-ADM1 model implementation, compared with the results obtained with ODE-ADM1 model** (*Allen et al., 2023*). Lines with circles correspond to values obtained in the current simulations. Curves with squares are obtained with the ODE-ADM1 model, showing virtually identical results.

that minimizes the error of the obtained methane production curve along with the COD at the beginning of the BMP assay, as seen in Fig. 2. Therefore, the proposed methodology, using the combined ADM1 model, solves the challenging problem of characterizing the influent sludge in a systematic way.

This methodology can be applied to support the development of experiments and full-scale reactor projects. The scalability and transferability of the data obtained with BMP assays can be used to apply the results in larger-scale systems (*Jingura & Kamusoko, 2017*).

The feasibility of using the model as a tool to predict BMP assays results and reduce possible mistrust in experimental results can be further evaluated by comparing the COD decrease observed experimentally in bench-scale and the values obtained in the numerical simulation. Also, the methanogenic activity and the production curve can be compared and calibrated to validate the model.

## Optimization of operational parameters using the combined AR-BMP model calibrated with BMP assays

Once the AR-BMP simulator is calibrated, it can be employed to optimize operational parameters of the AR system. As the new Julia language implementation of the BSM2 DAE ADM1 model developed in this work is very fast, when compared to previous implementations, it is very well suited for optimization purposes.

To demonstrate the capabilities of the methodology, an example of the optimization of the AR operational parameters is presented. In this example, the influence of the parameters
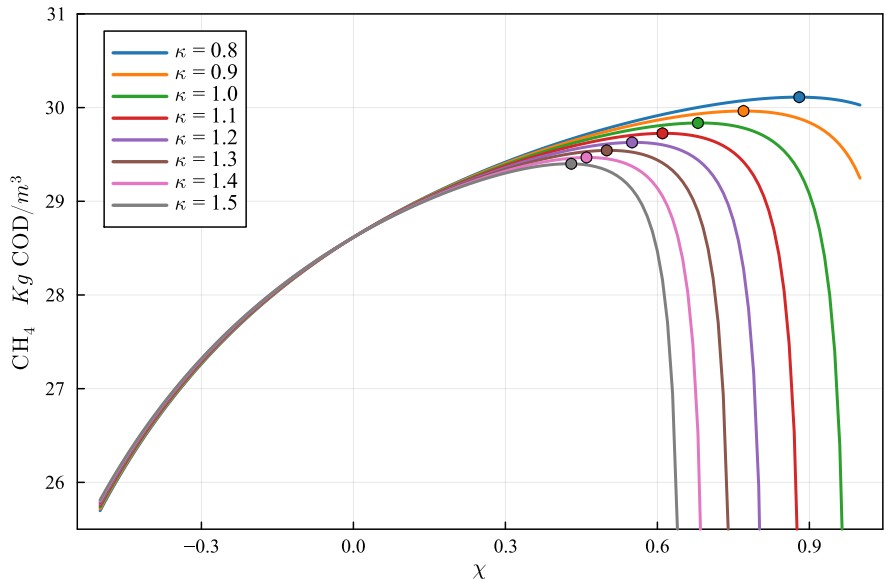

**Figure 11** **Optimization of the production curve of CH$_4$, considering a simple mixing/stratification model.** The optimum production for the case $\kappa = 1.0$ is obtained for $\chi = 0.68$, and the optimum production value is 29.84 kg COD/m³ of effluent. The optimum values, for different values of $\kappa$, is marked by a dot in each curve. The relevant data is shown in Table 9.

solute residence time, and particulate residence time is analyzed. The residence time can be increased by improving the flow patterns on the AR with inappropriate mixing, reducing slow recirculation flow regions or dead zones and dead time. On the other hand, particulate residence time can be selectively increased by providing a low mixing region at the outflow region, allowing for particle segregation due to the settling velocity.

To perform an optimization study, we define the following parameters:

$$\chi = \frac{V_p}{V} - 1$$

$$\kappa = \frac{V - V_s}{V_p - V}$$

where $V, V_p, V_s$ are the reactor nominal effective volume, the particulate effective volume, and the solute effective volume. Results of the simulations using the calibrated model for various values of $\chi$ and $\kappa$ are shown in Fig. 11. It can be observed that the CH$_4$ production improves, for all values of $\kappa$, with the increase of $\chi$ from 0 up to an optimal value. The optimization provides the best value of $\chi$ for each value of the parameter $\kappa$, keeping the same reactor nominal effective volume $V$. Table 10 shows the CH$_4$ production at the optimum $\chi$ for each of the simulated $\kappa$ values. It can be observed that CH$_4$ production can be increased by more than 5%, with respect to the unmodified parameter values, by optimizing the values of $\chi$ and $\kappa$.

**Table 10  Optimization of residence time parameters $\kappa$ and $\chi$ to improve CH$_4$ production.**

| $\kappa$ | Optimal $\chi$ | CH$_4$ production | Production increment (%) |
|---|---|---|---|
| 0.8 | 0.88 | 30.11 | 5.23 |
| 0.9 | 0.77 | 29.96 | 4.71 |
| 1.0 | 0.68 | 29.83 | 4.26 |
| 1.1 | 0.61 | 29.73 | 3.88 |
| 1.2 | 0.55 | 29.63 | 3.54 |
| 1.3 | 0.50 | 29.54 | 3.24 |
| 1.4 | 0.46 | 29.47 | 2.98 |
| 1.5 | 0.43 | 29.4 | 2.74 |

This methodology can be employed, along with numerical simulations of the particulate fluid flow or the results of properly reduced scale models, to assess the potential benefits of a proposed improvement of the process.

## CONCLUSIONS

The present study was based on the assembly of bench-scale bioreactors (BMP assays) and the use of experimental data obtained to feed the ADM1 mathematical model with subsequent calibration to simulate the anaerobic digestion of an anaerobic sludge obtained at a municipal WWTP. The bench scale BMP assays resulted in cumulative CH$_4$ production ranging from 124 to 143 NmL CH$_4$ and CH$_4$ yields ranging from 113 to 130 mL CH$_4$ g$^{-1}$ VS$^{-1}$ after 11 days of BMP assay.

A method that employs two coupled ADM1 model applications and BMP assay, combined with an iterative optimization method, was developed to characterize sewage sludge and biogas production potential by an equivalent composition that produces similar methanogenic curves.

The approach presented in this investigation can be used to design experiments in batch reactors using sewage sludge as feedstock to produce biogas, and to optimize the biogas production in the large scale biogas plant, which is very important for practical use.

Further studies applying BMP assays and computer simulations of both AR and BMP processes, including flow and particle transport simulations, are recommended for better understanding of the anaerobic digestion process. The results achieved can be used not only to define experiments in batch reactors having sewage sludge as feedstock to produce biogas but also to investigate microbial communities associated with methane production using molecular biology tools. Finally, the combination of BMP tests and biological data will be useful to predict the best conditions to operate anaerobic reactors to produce bioenergy.

## ACKNOWLEDGEMENTS

We acknowledge the constructive contributions from the reviewers that have improved the overall quality of this work.

### Funding

This research was supported by the Department of Innovation of the Rio de Janeiro State University (UERJ); the State Company of Water and Wastewater (CEDAE); the Carlos Chagas Filho Foundation for Supporting Research in the State of Rio de Janeiro (FAPERJ) (Proc. E-26/202.894/2018), the National Council for Scientific and Technological Development (CNPq) (Proc. 308335/2017-1) and the Brazilian Innovation Agency (Finep) (01.19.0087.00). The National Council for Scientific and Technological Development (CNPq) (Proc. 310955/2022-0) supported the APC. The funders had no role in study design, data collection and analysis, decision to publish, or preparation of the manuscript.

### Grant Disclosures

The following grant information was disclosed by the authors:
The Department of Innovation of the Rio de Janeiro State University (UERJ).
The State Company of Water and Wastewater (CEDAE).
The Carlos Chagas Filho Foundation for Supporting Research in the State of Rio de Janeiro (FAPERJ): Proc. E-26/202.894/2018.
The National Council for Scientific and Technological Development (CNPq): Proc. 308335/2017-1.
Brazilian Innovation Agency (Finep): 01.19.0087.00.
The National Council for Scientific and Technological Development (CNPq): Proc. 310955/2022-0.

### Competing Interests

Marcia Marques is an Academic Editor for PeerJ.

### Author Contributions

- Mariana Erthal Rocha conceived and designed the experiments, performed the experiments, analyzed the data, prepared figures and/or tables, authored or reviewed drafts of the article, and approved the final draft.
- Thais Carvalho Lazarino performed the experiments, authored or reviewed drafts of the article, and approved the final draft.
- Gabriel Oliveira performed the experiments, authored or reviewed drafts of the article, and approved the final draft.
- Lia Teixeira conceived and designed the experiments, analyzed the data, prepared figures and/or tables, authored or reviewed drafts of the article, and approved the final draft.
- Marcia Marques conceived and designed the experiments, analyzed the data, prepared figures and/or tables, authored or reviewed drafts of the article, and approved the final draft.

- Norberto Mangiavacchi conceived and designed the experiments, analyzed the data, prepared figures and/or tables, authored or reviewed drafts of the article, and approved the final draft.

## Data Availability

The raw data are available in the Supplemental File.

## Supplemental Information

Supplemental information for this article can be found online at http://dx.doi.org/10.7717/peerj.16720#supplemental-information.

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
