# Peer review of "Analysis of biogas production from sewage sludge combining BMP experimental assays and the ADM1 model"

_PeerJ, doi:10.7717/peerj.16720_

## Round 0.1 · original submission · Major Revisions

Please address all the comments provided by the reviewers. The first reviewer might have not understood the simulation graphs you provided; therefore, please make a better work in your description. Comments 1, 2, and 3 by the second reviewer must be addressed to have a better shot at publication.

Reviewer 1 ·

Basic reporting

This article has many flaws, both in terms of content and structure, which justifies its rejection. Here are some considerations:
- The title of the work refers to the modeling of biogas production, however the application of an already developed model (ADM1) was carried out, not the structuring of a model.
- Two ADM1 models were not used to simulate methane production, the model was applied to simulate two different systems.
- No results were presented for variables calibrated by the model, nor simulation graphs, statistical analyzes and model validations to characterize a modeling work
- The tables are incompatible with the descriptions present in the text and the captions are not self-explanatory
- The figures do not have a good quality and also do not have explanatory captions
- The ADM1 model was not developed in 2020 by Donoso-Brava as mentioned
- The parameters were taken from the original ADM1, but which ones were not highlighted and, mainly, which were the differences in both works.
- The work needs to be better structured and English improved.

Experimental design

The experimental design is not clear and the modeling part was not described properly.

Validity of the findings

The results were not well presented in the graphs and tables and the discussion is very superficial.

·

Basic reporting

In principle the objective of the study is interesting and of practical importance. The structure of the article conforms to the standards of PeerJ. The figures and tables are relevant. The literature seems to be well referenced and relevant.
Furthermore, there are a number of errors that should be corrected. The text contains a number of mistakes and missing blank spaces (e. g. between value and unit). In the final analysis, the reviewed article seems to be suitable for publication after revision.
The specific comments are summarized in the attached pdf file “Specific comments_peerj-85542”.

Experimental design

No comment.

Validity of the findings

No comment.

Additional comments

The following issues should be added/discussed (sorted by importance):
(1) Lines 380 – 384: From my point of view it would make more sense to use the model for the BMP assay to determine and calibrate the model parameters for the second model (bioreactor). The optimization of the biogas production in the large scale biogas plant is much more important for practical use.
(2) Where all model parameters calculated by the mathematical optimization tool/numerical calculation? The procedure of the numerical calculation should be explained in more detail.
(3) Table 5: The values have to be discussed/explained (concentration of inerts Xi and Si is higher than the concentration of the composite Xc, concentration of the biomass fractions Xaa and Xac is very high compared to the other 5 biomass fractions). Where these values determined experimentally or by numerical calculations?
(4) Lines 369 – 371: The authors should consider the model ADM1xp (This model contains a new fraction Xp for particulate inerts from biomass decay). [Wett, B., A. Eladawy und M. Ogurek (2006). „Description of nitrogen incorporation and release in ADM1“. In: Water Science and Technology 54.4, S. 67–76. issn: 0273-1223. doi: 10.2166/wst.2006.527.]
(5) Lines 372 – 374: The authors should also consider Weende analysis and van Soest extension for characterization of sludge/substrate (these methods have already been used in numerous studies for the characterization of substrates). [Naumann, C. und R. Bassler (1993). Die chemische Untersuchung von Futtermitteln. Bd. 3. Methodenbuch. Darmstadt, Deutschland: VDLUFA-Verlag. isbn: 9783941273146.] [van Soest, P. J., J. B. Robertson und B. A. Lewis (1991). „Methods for dietary fiber, neutral detergent fiber, and nonstarch polysaccharides in relation to animal nutrition“. In: Journal of Dairy Science 74.10, S. 3583–3597. issn: 00220302. doi: 10.3168/jds.S0022-0302(91)78551-2.] [van Soest, P. J. und R. H. Wine (1967). „Use of detergents in the analysis of fibrous feeds. IV. Determination of plant cell-wall constituents“. In: Journal of Association of Official Analytical Chemists 50.1, S. 50–59. doi: 10.1093/jaoac/50.1.50.]
(6) Line 312: The statement „CH4 biogas“ should be clarified (biogas consists not only of methane).
(7) The statement in lines 314/315 is a repetition of line 313.
(8) The statements in line 352 „each kilogram of COD removed will yield 0.35 m3 of biogas“ and lines 354/355 „CH4 production of approximately 0.35 L g-1 of COD removed“ do not appear to be consistent and have to be clarified (see above, (6))
(9) Figure 1: Description of the different parts of the experimental setup is missing.
(10) Figure 4: “R1” appears twice, “R2” is missing.

---

## Round 0.2 · accepted · Accept

According to the review, the authors have addressed all the comments.

·

Basic reporting

In my view, the authors have implemented all the comments made by the reviewers and have discussed the issues raised in a well-founded manner and made appropriate changes to the text. In addition, all suggested corrections were made and further important content/aspects were included in the article. I therefore consider the current version of the text to be suitable for publication in the current version.

Experimental design

No further comments (see point 1).

Validity of the findings

No further comments (see point 1).

Additional comments

No further comments (see point 1).